# Vegan Ice Cream Made from Soy Extract, Soy Kefir and Jaboticaba Peel: Antioxidant Capacity and Sensory Profile

**DOI:** 10.3390/foods11193148

**Published:** 2022-10-10

**Authors:** Giovana M. N. Mendonça, Estela M. D. Oliveira, Alessandro O. Rios, Carlos H. Pagno, Daniela C. U. Cavallini

**Affiliations:** 1São Paulo State University (UNESP), School of Pharmaceutical Sciences, Araraquara, São Paulo 01049-010, Brazil; 2Institute of Science and Food Technology, Federal University of Rio Grande do Sul, UFRGS—Rio Grande do Sul, Porto Alegre 90010-150, Brazil

**Keywords:** probiotic, ice cream, soy extract

## Abstract

Considering the need for functional foods and the use of by-products of the food industry, a potentially functional ice cream was developed, using soy extract, soy kefir and dehydrated jaboticaba peel. Five ice creams were produced using soy kefir (K) and soy extract (S): (1) GS—100% S; (2) GK1-75% S/25% K; (3) GK2-50% S/50% K; (4) GK3-25% S/75% K and (5) GK-100% K; The products were evaluated by physicochemical, microbiological and sensory (check all that apply) analyses. The addition of kefir was found to increase the acidity of the products. The concentrations of total phenolic compounds in the formulations with kefir were approximately ten times higher than the GS formulation. All products presented concentrations of thermotolerant coliforms <3 NMP/g and absence of *Salmonella* ssp. The viability of *Lactobacillus* ssp., *Streptococcus* spp. and *Bifidobacterium* ssp. was higher than 10 log CFU/g during the whole storage period. The GS and GK1 formulations had the lowest scores, while GK ice cream was preferred. The formulations showed distinct sensory profiles in the CATA, and the ice cream with 100% kefir was associated with desirable attributes. The ice creams exhibited microbiological and sensory characteristics that meet the expectations of the product’s target audience.

## 1. Introduction

Consumer demand for a healthy and balanced diet has driven the food industry to seek alternatives that can meet this demand. Among the products developed focusing on this market are those with reduced trans-fat and sodium content, whole foods and organic, vegan and functional foods, including probiotics [1].

Kefir is defined as a “fermented milk produced by inoculating kefir grains or starter culture, composed of *Lactobacillus kefir*, species of the genera *Leuconostoc* spp., *Lactococcus* spp. and *Acetobacter* spp. and lactose-positive and/or lactose-negative yeasts, which grow synergistically” [2]. This beverage is recognized worldwide as an excellent source of microorganisms, which have potential health benefits. The microorganisms present in kefir and the products of their symbiotic relationship are related to antimicrobial, anti-inflammatory and anti-allergic effects attributed to the beverage [3,4,5,6].

Cow’s milk kefir is the most popular beverage, but its consumption is limited for lactose-intolerant, dairy-allergic and vegan people, indicating the need to adapt the culture to non-dairy substrates, especially those obtained from cereals and legumes [7].

The water-soluble soy extract can be used as a substitute for cow’s milk, due to its good nutritional profile and lactose-free characteristic, for use in products intended for consumers that are lactose-intolerant or allergic to milk protein [8]. The soy extract is mainly consumed as a ready-to-drink beverage or as an ingredient used in various formulations such as flavored drinks, fermented drinks and desserts, including ice cream [9].

Ice cream is a frozen product obtained from an emulsion of fats and proteins or a mixture of water and sugar, allowing the addition of other ingredients, which does not de-characterize the product. This frozen dessert is widely accepted by consumers, with a worldwide consumption per capita of 5.29 L [10,11].

Due to changes in the population’s eating habits, the market for vegetable-based ice creams has expanded, with the incorporation of fruits with functional components, such as carotenoids, fibers and phenolic compounds.

Among Brazilian fruits, the jaboticaba stands out for having a sweet and slightly spicy flavor and dark purple peel, rich in minerals, vitamin C, soluble fiber and phenolic compounds (especially anthocyanins). Such compounds can be used as a natural dye and have antioxidant properties, with potential beneficial effects on health, such as modulation of the lipid profile and anti-inflammatory and anti-atherogenic properties [12].

The modification of the raw material and ingredients used in the preparation of ice cream can affect the textural properties, flavor, color and product acceptance. Thus, the aim of this work was to develop a potentially functional ice cream, from soy extract, soy kefir and dehydrated jaboticaba peel, which can be consumed by lactose-intolerant, milk-protein-allergic and vegan individuals.

## 2. Materials and Methods

### 2.1. Material

#### 2.1.1. Obtaining Soy Kefir

The soy extract was obtained from the Soy Derivatives Production and Development Unit (Universoja—FCF—UNESP Araraquara) [13]. The soy extract was fermented by a mixed starter culture (CHOOZIT Kefir DC LYO 1000 L), kindly provided by DANISCO (DuPont, Paulinia—SP, Brazil). After the addition of sucrose (10 g per liter), the soy extract was heated to 95 ± 3 °C for 30 min, cooled to 25 ± 2 °C and inoculated with the commercial starter culture (0.005 g/L). The mixture was incubated at 37 ± 2 °C until a pH of 4.80 was reached.

#### 2.1.2. Production of Dehydrated Jaboticaba Peel Flour

The ripe fruits were purchased from local producers of Araraquara-SP in August and October 2018, the jaboticaba’s harvest period. The fruits were selected, sanitized in an aqueous sodium chloride solution (100 mg/L) for 30 min and rinsed in running water. The residue was dehydrated in ovens equipped with air circulation at 60 °C to constant moisture content (12.05 ± 0.27 g/100 g). Subsequently, the dehydrated residue was ground in a food processor, placed in glass jars covered with aluminum foil and stored at room temperature until use.

#### 2.1.3. Production of the Ice Cream

Five ice creams were produced, with different concentrations of soy kefir and soy extract Table 1. The following ingredients were used (g/100 g wet basis): 1.3 g of glucose, 1.0 g of emulsifier (Emustab), 0.80 g of dehydrated jaboticaba peel and 5.0 g of condensed soy milk. After homogenizing the ingredients for five minutes, the mixture was left to rest for 24 h, which corresponds to the maturation period. Then it was frozen and air incorporated (overrun) in a ice cream maker (MDG, model MH 80/100—São Carlos- Brazil).

### 2.2. Methods

The formulations were processed on two separate occasions and evaluated during a 90-day storage period at −24 ± 2 °C. The proximate composition of the ice creams was determined in the freshly processed product, and the microbiological safety and viability of the potentially probiotic microorganisms were monitored at 15-day intervals. The other parameters (color, phenolic compounds, antioxidant activity, anthocyanin concentration and sensory profile) were evaluated immediately after preparation (T0) and at the end of the storage period (T90). The physicochemical and microbiological analyses were performed in triplicate.

#### 2.2.1. Microbiological Analyses

Serial decimal dilutions were performed up to 10^−8^ (25 g of ice cream in 225 mL of sterile 0.1% peptone water). Microbiological quality was evaluated by the enumeration of coliforms at 45 °C, using the most probable number (MPN) assay [14] and detection of *Salmonella* spp. [15]. The population of potentially probiotic microorganisms was performed by plating on specific culture media: *Lactobacillus* spp.—Lactobacilli Man Rogosa Sharpe ágar—MRS (Difco, França), *Streptococcus* spp.—M17 ágar and *Bifidobacterium* spp.—BIM-25 (Reinforced Clostridium Ágar—Difco, França—with the addition of nalidixic acid, polymyxin B sulfate, kanamycin sulfate, iodoacetic acid and triphenyl tetrazolium chloride), respectively—using the microdroplet technique [16]. MRS and M17 plates were incubated in aerobiosis at 37 °C/48 h and BIM-25 plates in anaerobiosis at 37 °C/72 h [17]. Yeast enumeration was performed by surface plating on the Yeast Malt medium (YM, Himedia, India) with added chloramphenicol (200 mg/L) and incubation at 22 °C/72–120 h. The results were expressed as colony-forming units per gram (CFU/g).

#### 2.2.2. Physicochemical Analyses

##### Proximate Composition, Caloric Value, pH and Titratable Acidity

The moisture, ash, protein and lipid contents were determined according to the Analytical Standards of the Adolfo Lutz Institute [18]. Total carbohydrate content was determined by difference [19]. The caloric value of the ice cream was estimated from the carbohydrate (4 kcal/g), protein (4 kcal/g) and fat (9 kcal/g) contents, and the result was expressed in kcal per 100 g. The pH was determined by a potentiometric method using a digital pH meter and the titratable acidity by titration with 0.1 N sodium hydroxide solution and expressed as % lactic acid [18].

##### Total Phenolic Compounds and Total Antioxidant Activity

The ice cream sample for the determination of phenolic compounds, antioxidant activity and anthocyanin were freeze-dried to preserve their characteristics in the evaluated time periods. The extracts for the determination of total phenolic compounds and antioxidant activity were obtained according to the procedure described by Karaaslan et al. [20]. The quantification of total phenolic compounds was performed by the Folin–Ciocalteu method [21]. Briefly, in a volumetric flask, the ice cream sample (0.1 mL), Folin–Ciocalteau reagent (0.5 mL), 20% sodium carbonate solution (1.5 mL) and distilled water were added to obtain a final volume of 10 mL. The mixture was kept at rest for two hours, and the absorbance was read at room temperature in a spectrophotometer (Shimadzu UV-Vis 1800/08302) at 765 nm. The results were expressed as milligrams of gallic acid equivalents (mgGAE) per 100 g of sample.

The antioxidant capacity was determined by capturing the ABTS free radical reaction according to the application used by Rufino et al. [22]. For the extract preparation, 20 mL of 50% methanol was added to one gram of the sample, followed by homogenization in Ultra-Turrax (IKA^®^/T25 digital, Staufen, Germany) and incubation for 60 min. The extract was centrifuged (25,400 G/15 min) and the supernatant was stored in an amber flask. Then, the operation was repeated using 70% acetone. In a place without light, the obtained extracts were diluted, and a 100 μL aliquot of each dilution was mixed with 1.0 mL of the ABTS radical. After homogenization, the samples were left to stand for six minutes, and an absorbance reading was performed at 734 nm in a spectrophotometer (Shimadzu UV-1800). The analyses were performed in triplicate and the results are expressed in μmol Trolox/g.

##### Anthocyanin’s Quantification

For the extraction of anthocyanins, 0.5 g of freeze-dried ice cream samples were diluted (1% HCL methanolic solution), homogenized (Ultra-Turrax^®^) [23] and vacuum filtered (Millex LCR 0.45 μm, 13 mm filter) [24]. The extractions were repeated until the color of the samples disappeared, and the obtained extracts were stored at 7 ± 1 °C for further analysis. The identification and quantification of anthocyanins were carried out by HPLC [25], using a chromatograph (Agilent^®^, Series 1100, Santa Clara, CA, USA) equipped with a quaternary pump system and UV-visible detector. A C18 Shim-pak CLC-ODS reverse-phase column (5 μm, 250 × 4.6 mm) was used for the separation of anthocyanins, and the mobile phase consisted of a linear elution gradient of formic acid (5%) and methanol, respecting the ratio of 85:15 (*v*/*v*) to 20:80 over 25 min, maintained for 15 min. The flow rate of the mobile phase, the injected volume and the column temperature used were 1.0 mL·min^−1^, 5 μL and 29 °C, respectively. The chromatograms were processed at 520 nm, and standard curves—constructed from standards of cyanidin 3-glycoside and delphinidin 3-glycoside (Sigma-Aldrich, St. Louis, MI, USA)—were used for the quantification.

##### Instrumental Color Determination

The color parameters of the samples were determined in a Konica Minolta portable colorimeter (CR-410) using illuminant D65 and 10° visual angle—using the CIEL L * a * b * system—whereas the chroma and hue values were calculated based on parameters a * and b * [22].

##### Sensory Analysis

A total of 115 consumers (33 men and 82 women, aged between 18 and 59 years old) participated in the acceptance [26] and Check All That Apply tests (CATA). Participants were recruited from students and staff at the School of Pharmaceutical Sciences at UNESP—Araraquara, through questionnaires applied to assess allergies, intolerances or diseases that prevented participation in the tests and to identify aversion to any ingredient of the ice cream. The samples were labeled with 3-digit codes and presented monadically in random order. Data collection took place at the Sensory Laboratory of the School of Pharmaceutical Sciences in sensory booths with controlled temperature (23 ± 2 °C) and white lighting. The sensory analysis was approved by the Research Ethics Committee (REC 003167/2019).

The attributes of appearance, aroma and flavor were evaluated in the acceptance test using a nine-point structured hedonic scale, ranging from “I liked it a lot” to “I disliked it a lot” [27]. In the purchase-intention test, a five-point nominal category scale was used, ranging from “would definitely buy the product” to “would definitely not buy the product” [28].

Consumers were instructed to fill out a CATA questionnaire, containing attributes or phrases describing the ice cream samples [29]. Such attributes or phrases were previously generated using the grid method by ten trained assessors habituated to eating soy-based products and performing descriptive testing. The final list of attributes was defined in consensus with the assessor’s team [30]. Consumers were instructed to point out any attributes they considered suitable to describe the ice cream samples.

#### 2.2.3. Statistical Analysis

The data from microbiological and physicochemical tests were evaluated by analysis of variance and Tukey’s test of means or a *t* test, (*p* < 0.05). The acceptance test results were analyzed by analysis of variance, considering the sample and consumers as sources of variation, and Tukey’s test of means (*p* < 0.05). For the CATA, the frequency of indication of each term was determined, and the Cochran Q test was applied to identify significant differences among samples for each of the sensory attributes. Furthermore, Multiple Correspondence Analysis (MCA) was performed on the frequency table, containing the samples in rows and the CATA questionnaire terms in columns.

## 3. Results and Discussion

### 3.1. Microbiological Analyses

GS, GK1, GK2, GK3 and GK ice creams showed the absence of *Salmonella* spp. and had coliform counts at 45 °C < 3 NMP/g during the storage period (90 days), being considered safe for human consumption [14]. Regarding the potentially probiotic microorganisms, it was not possible to detect the presence of yeasts in the ice creams after 0, 15 and 30 days of storage at −22 °C ± 2. However, after 45 days of storage, the yeast count was above 8 log CFU/g for all the formulations, indicating a possible adaptation to the product conditions. The population of *Bifidobacterium* spp., *Lactobacillus* spp. and *Streptococcus* spp. was higher than 10 log CFU/g in all formulations throughout the storage period. Conventionally, kefir is prepared with cow’s milk, an ideal medium for lactic fermentation. In this study, replacing cow’s milk with soy extract did not reduce the viability of potentially probiotic bacteria, indicating an adaptation of the starter culture to the ice cream ingredients and processing conditions (Table 2). Santos et al. [31] also observed high counts of *Lactococcus* spp., *Lactobacillus* spp. and yeasts in soymilk kefir with prebiotic addition, after 28 days of storage. Likewise, Walter et al. [32] verified that *Bifidobacterium animalis* subsp. *lactis* BB-12 and *Lactobacillus acidophillus* LA-5 populations remained stable during the storage period using different soy-based raw materials (soybean of the BRS 232 cultivar and commercial soy powder extracts obtained by liquid extraction and solid extraction) to produce kefir.

Ice cream’s processing steps, such as homogenization, churning and the consequent incorporation of oxygen, as well as low storage temperatures, may result in decreasing the population of beneficial microorganisms. Temperature fluctuations during storage, resulting in the formation of ice crystals, can also reduce the survival of strains [32]. However, in the present work, the bacteria and yeast from the kefir starter culture were resistant to the stress conditions of the process, requiring no additional strategies to protect them.

There is no consensus about an appropriate dose of probiotic organisms required to achieve beneficial effects. According to the International Scientific Association for Probiotics and Prebiotics (ISAPP), all developed ice creams should be designated as a product “containing live and active cultures”. These products could not be considered a probiotic since they have a diverse microbial community that is not completely defined, in terms of the composition and stability of the strains, and without proof of beneficial effects. All ice cream also showed a minimum of 9 log CFU per serving, as suggested by ISAPP [33].

### 3.2. Physicochemical Analyses

The results of the physicochemical analyses are presented in Table 3. Overall, kefir caused a decrease in the pH values and an increase in the titratable acidity of the ice creams. This effect was expected since kefir is a product fermented by lactic and acetic bacteria, which metabolizes carbohydrates and produces acids, mainly lactic, acetic, citric, propionic and butyric acids [34,35].

Kefir also caused a significant decrease in the moisture content of the ice cream (moisture GK = 69.61 ± 0.33 g/100 g and GS = 80.25% ± 0.14 g/100 g), and this result was expected because kefir has a lower moisture content than soy extract (Kefir: 84.47 ± 0.18 g/100 g; soy extract 94.25 ± 0.08 g/100 g). The GK formulation (100% kefir) showed the highest carbohydrate content (21.59 ± 0.12 g/100 g), differing from the other formulations produced with a mixture of kefir and soy extract. This behavior is explained by the addition of sugar (10% *w*/*v*) to the soy extract used to obtain the soy kefir. The GK and GS formulations have the highest and lowest caloric values, with 139.52 and 98.02 Kcal/100 g, respectively, because of their carbohydrate contents. The protein and lipid contents were similar among all formulations.

### 3.3. Total Phenolic Compounds and Antioxidant Capacity

In T0, formulation GS presented the lowest content of total phenolic compounds (567.65 ± 35.60 mgGAE/100 g, *p* < 0.05) while GK had the highest average of such compounds (7631.69 ± 47.73 mgGAE/100 g, *p* < 0.05). Higher stability in total phenolic content was observed for GK3, GK and GK2 formulations with a reduction of only 6.94%, 7.73% and 8.29%, respectively, after 90 days of storage. However, the opposite effect was observed in the GS formulation with a reduction of 52.26% (Table 4).

The concentrations of total phenolics in each ice cream formulation were higher than those found by other authors who used jaboticaba in the processing of symbiotic concentrated yogurt with 1% of jaboticaba bark flour (292.5 ± 2.5 mg GAE/100 g) [36] and jaboticaba juice (150.4 ± 0.6 mg/L) [37]. The bark of jaboticaba is rich in phenolic compounds, such as gallic and ellagic acid, rutin and quercetin [38,39], and soy extract is a source of these same compounds and isoflavones, ferulic, gallic and vanillic acids [40], which justifies the high levels of total phenolics in ice cream. The higher concentration of total phenolic compounds in the samples containing kefir is likely due to the metabolic activity of the starter culture. Enzymes derived from microorganisms—such as ß-glycosidase—hydrolyze complex phenolic compounds into simpler ones, leading to an increase in the total phenolic content [41,42].

The ice creams processed with kefir also exhibited the highest antioxidant capacity, without differing from each other. However, after 90 days of storage, only the sample without kefir (GS) did not show a significant reduction in this parameter. This result cannot be attributed to the concentration of total phenolic compounds because the formulations with kefir showed a lower reduction in these compounds at the end of the storage period. The antioxidant activity of a food is related to the concentration and chemical structure of bioactive compounds (phenolic compounds, vitamins and enzymes superoxide dismutase, catalase and peroxidase), which may change during storage. In addition, synergy among compounds may result in increased antioxidant activity [43,44] and explain the observed difference.

### 3.4. Anthocyanin Content

The concentration of anthocyanins was expressed as cyanidin 3-glucoside and delphinidin 3-glucoside (Table 5), since these compounds are found in higher concentrations in jaboticaba bark and are related to its purple color.

The concentration of anthocyanins in ice cream formulations is related to the amount of peel added (0.02%), which was identical for all formulations. Delphinidin was the most abundant anthocyanin, contrary to the data obtained by Reynertson et al. [39] and Inada et al. [37] who reported the predominance of cyanidin in jaboticaba and jaboticaba juice, respectively. The formulation without kefir (GS) presented the highest values of cyanidin 3-glycoside and delphinidin 3-glycoside, differing significantly from the other formulations at the two evaluated times. The lower values of anthocyanins found in samples GK, GK1 and GK2 may be related to the activity of kefir strains. Some microorganisms can produce enzymes that hydrolyze anthocyanins into less stable aglycone forms and/or can produce hydrogen peroxide that facilitates their degradation [45]. A significant reduction in the concentration of anthocyanins was observed during storage for all formulations. Such behavior was expected, since anthocyanins are unstable to variations of temperature, oxygen, light, pH and acidity of the medium [46,47]. The concentration of anthocyanins can also be affected by a mixture of copper, iron and manganese, which act as catalysts for the oxidation reaction and are found in the bark of jaboticaba [48].

One of the challenges of using anthocyanins as natural dyes in foods is their instability in the face of pH variations, which affect the color conferred by these compounds. While a low pH is associated with greater stability of the compound, an increase in the pH results in a reduction in intensity and a change in the color pattern conferred [49]. The pH of the ice cream (Table 3) did not interfere positively with the concentration and stability of anthocyanins (Table 5), and we can infer that the different processing steps, which lead to the incorporation of oxygen in the samples, may have influenced the results obtained.

### 3.5. Color Determination

Table 6 shows the results for the color parameters (L *, a *, b *, C * and H°) during storage (T0 and T90). Briefly, during the storage period, the samples produced with kefir became darker (L * reduction) and with lower color intensity (C * reduction). The red color was predominant in all samples, except for GK1.

The GK and GK3 formulations were the lightest samples showing the highest averages for parameter L * at T0 and T90. The values of a * were positive for all formulations, indicating a predominance of red color, and only GK1 showed a significant reduction in this parameter at the end of the storage time. The b * value was also positive for all formulations, with a tendency of a yellow color.

Chroma (C *) is a quantitative parameter related to the color intensity perceived by the human eye. The GK1, GK3 and GK samples have the highest averages for C * at T0, and the formulation with 50% kefir (GK2) exhibited the greatest reduction in this parameter at the end of 90 days. The results of the hue angle (h°—qualitative attribute of the color) indicate that GK (T0 and T90), GK3 (T0) and GK2 (T90) are characterized by a reddish color, while in the other samples, there was a predominance of yellow.

The characteristic coloration of the anthocyanin pigment is variable and can give the product a pink, red or blue color, depending on the source. Although equal concentrations of dehydrated jaboticaba peel were used, the addition of different concentrations of kefir may interfere with the color of the product, as well as with its stability. In summary, the results indicate that the ice creams are characterized as dark, with a color ranging from red to yellow and with low saturation. GK1 and GK2 exhibited the lowest color stability during storage, and the one prepared with 100% kefir (GK) showed a tendency toward the most pronounced red color.3.6. Sensory Analysis.

In T0, GK and GK2 showed the highest acceptance averages for the flavor attribute, without differing from GK3 and GS. There were no significant differences (*p* > 0.05) among the ice creams in appearance, color and aroma. After 90 days of storage, GK showed the highest hedonic averages for the sensory attributes evaluated, differing from all formulations regarding flavor. It is also noteworthy that the sample with 100% kefir (GK) was the only one that did not show a reduction in color acceptance during the storage time (Table 7).

Formulations GK, GK3 and GK2 showed the highest frequencies of positive purchase intention (“would certainly or probably buy the products”) at T0 (44.2%, 38.4% and 34.2%) and T90 (59.8%, 34.5% and 31.1%) (Table 7). On the other hand, the highest frequencies of negative purchase intention (“probably or certainly would not buy”) were obtained by GS (35.8% at T0 and 35.2% at T90) and GK1 samples (42.5% at T0 and 47.5% at T90). These results agree with those obtained in the hedonic test for storage time T0, where formulations GS and GK1 exhibited the lowest acceptance averages for all the attributes evaluated and GK was the most accepted.

A total of 24 attributes were previously selected by the trained assessors to characterize the ice cream by the CATA method. The frequencies of the attributes and the results of Cochran’s Q-test for the freshly processed product (T0) are reported in Table 8. The most selected attributes were soy flavor, aftertaste, refreshing flavor, natural jaboticaba flavor and sweetness. Cochran’s Q test showed significant differences in 6 of the 24 attributes analyzed, and all of them were related to flavor (sweet, jaboticaba flavor, acid taste, refreshing flavor, soy flavor and yogurt flavor), indicating its importance for detecting differences and characterizing the ice cream.

After 90 days of storage, the CATA test was applied again to identify possible sensory changes as a function of time. The results of Cochran’s Q test showed significant differences (*p* < 0.05) in 14 of the attributes analyzed (Table 9). The attributes sweet, creamy texture, soy flavor, natural jaboticaba flavor, roughness, purple fruit color (berries), refreshing flavor and attractive color were mentioned more frequently, showing a change in the sensory profile of the formulations during the storage period.

Multiple correspondence analyses showed that the first and second dimensions explain 92.86% of the variance of the experiment at T0 (Figure 1a). The statistical test uses only the most frequently cited attributes since those with a low frequency can lead to false results [50]. The formulations were positioned in three different groups and obtained good attribute separation among the three quadrants. GS and GK1 were characterized by an aftertaste and a soy flavor, GK3 by a refreshing flavor and yogurt flavor, and GK2 did not obtain any relationship with the descriptor terms. The GK formulation, which exhibited the highest absolute mean for the flavor attribute, was characterized by a sweet and natural jaboticaba flavor, evidencing the impact of these characteristics on the product consumer. At T90 (Figure 1b), the first and second dimensions explained 94.62% of the variation between samples. The ice creams were positioned in four different quadrants and characterized by different attributes. The attribute that best described the GS formulation was a creamy texture; GK1 and GK2 were characterized by a purple fruit color (berries) and GK3 by a fermented flavor. Once again, the formulation with 100% kefir (GK) presented a distinct sensory profile, being associated with pleasant attributes such as a fruit aroma, refreshing flavor, natural jaboticaba flavor, attractive color and yogurt flavor.

The change in the descriptive profile generated by CATA after 90 days of storage may be associated with physicochemical characteristics of color and anthocyanin content, since color-related sensory attributes were not considered important in the freshly processed product. The color of foods and beverages has been found to play important roles in consumer perception of other sensory attributes (taste and aroma), product acceptance and emotional responses [51]. In this study, the results of instrumental color showed that the ice cream produced with 100% kefir (GK) had the highest intensity of red color, which likely contributed to the better sensory profile of this formulation.

## 4. Conclusions

The results indicated that these ice creams, especially the one prepared with 100% kefir (GK), are suitable vehicles for potential probiotic strains. The jaboticaba peels imparted a purple color and improved the antioxidant capacity and anthocyanin profile of the products. The use of alternative raw materials and ingredients made it possible to obtain a product with functional potential that can be ingested by individuals who adopt a diet with the restriction of lactose, milk protein or foods of animal origin.

## Figures and Tables

**Figure 1 foods-11-03148-f001:**
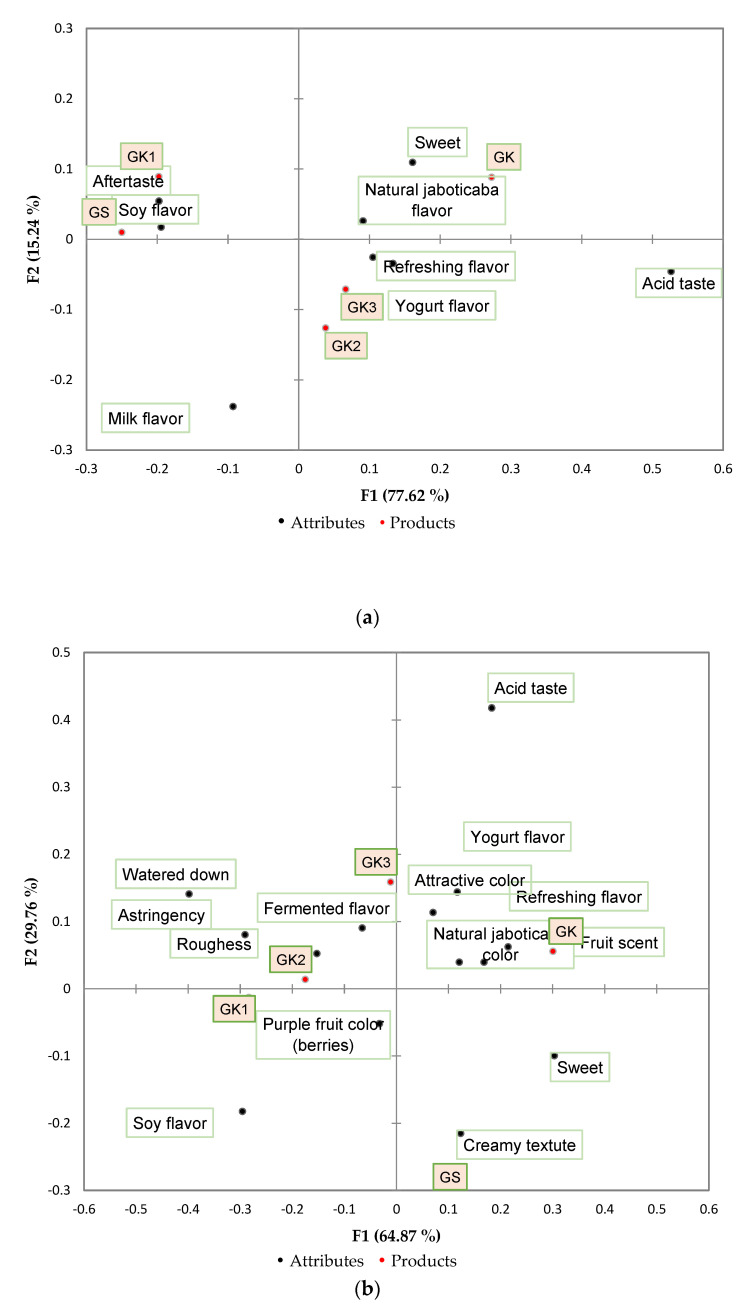
Correspondence analysis of the sensory descriptors in the CATA analysis for the initial time (T0) (**a**) and final time (T90) (**b**).

**Table 1 foods-11-03148-t001:** Formulations with different concentrations of soy extract (GS) and soy kefir (GK) (g/100 g wet basis).

Ingredients	Formulations (g/100 g Wet Basis)				
	GS	GK1	GK2	GK3	GK
Soy extract	100.0	75.0	50.0	25.0	-
Kefir	-	25.0	50.0	75.0	100.0

**Table 2 foods-11-03148-t002:** Viability of yeasts, *Bifidobacterium* spp., *Lactobacillus* spp. and *Streptococcus* spp. in ice cream during storage at −22 °C ± 2.

Formulations	Time (Days)	Yeasts Log UFC/g	*Bifidobacterium* spp. LogUFC/g	*Lactobacillus* spp. Log UFC/g	*Streptococcus* spp. Log UFC/g
GK1	0	<1	10.46 ^abc^ ± 0.07	10.60 ^a^ ± 0.07	10.79 ^a^ ± 0.02
	15	<1	10.38 ^bc^ ± 0.04	10.49 ^ab^ ± 0.02	10.76 ^ab^ ± 0.02
	30	<1	10.32 ^c^ ± 0.04	10.32 ^bc^ ± 0.02	10.73 ^abc^ ± 0.01
	45	8.93 ^a^ ± 0.20	10.31 ^c^ ± 0.02	10.35 ^bc^ ± 0.07	10.69 ^bcd^ ± 0.02 ^b^
	60	8.72 ^a^ ± 0.20	10.48 ^ab^ ± 0.04	10.42 ^ab^ ± 0.12	10.65 ^cde^ ± 0.03
	75	8.30 ^a^ ± 0.00	10.55 ^a^ ± 0.05	10.30 ^bc^ ± 0.07	10.62 ^de^ ± 0.03
	90	8.30 ^a^ ± 0.00	10.49 ^ab^ ± 0.10	10.21 ^c^ ± 0.08	10.57 ^e^ ± 0.04
GK2	0	<1	10.58 ^a^ ± 0.02	10.57 ^a^ ± 0.06	10.77 ^a^ ± 0.01
	15	<1	10.49 ^ab^ ± 0.05	10.53 ^ab^ ± 0.07	10.74 ^ab^ ± 0.01
	30	<1	10.37 ^bc^ ± 0.08	10.41 ^abc^ ± 0.11	10.71 ^abc^ ± 0.02
	45	9.77 ^a^ ± 0.21	10.34 ^bc^ ± 0.07	10.26 ^bcd^ ± 0.10	10.68 ^bcd^ ± 0.03
	60	9.36 ^b^ ± 0.10	10.41 ^abc^ ± 0.10	10.29 ^abcd^ ± 0.01	10.64 ^cde^ ± 0.04
	75	9.60 ^a^ ± 0.00	10.34 ^bc^ ± 0.05	10.15 ^cd^ ± 0.14	10.60 ^de^ ± 0.04
	90	9.30 ^b^ ± 0.00	10.29 ^c^ ± 0.01	10.03 ^d^ ± 0.14	10.56 ^e^ ± 0.04
GK3	0	<1	10.52 ^a^ ± 0.07	10.55 ^a^ ± 0.11	10.75 ^a^ ± 0.02
	15	<1	10.38 ^a^ ± 0.11	10.47 ^ab^ ± 0.02	10.71 ^ab^ ± 0.02
	30	<1	10.32 ^a^ ± 0.07	10.31 ^bc^ ± 0.02	10.68 ^abc^ ± 0.03
	45	9.92 ^a^ ± 0.08	10.40 ^a^ ± 0.09	10.31 ^bc^ ± 0.02	10.64 ^bcd^ ± 0.03
	60	9.70 ^ab^ ± 0.17	10.48 ^a^ ± 0.05	10.27 ^bc^ ± 0.04	10.60 ^cd^ ± 0.03
	75	9.36 ^b^ ± 0.10	10.41 ^a^ ± 0.05	10.14 ^cd^ ± 0.12	10.56 ^de^ ± 0.03
	90	9.40 ^b^ ± 0.17	10.35 ^a^ ± 0.06	10.06 ^d^ ± 0.10	10.50 ^e^ ± 0.04
GK	0	<1	10.48 ^a^ ± 0.09	10.57 ^a^ ± 0.08	10.79 ^a^ ± 0.03
	15	<1	10.35 ^ab^ ± 0.08	10.46 ^ab^ ± 0.09	10.77 ^ab^ ± 0.03
	30	<1	10.29 ^b^ ± 0.01	10.38 ^ab^ ± 0.11	10.74 ^ab^ ± 0.03
	45	9.77 ^a^ ± 0.21	10.34 ^ab^ ± 0.07	10.37 ^ab^ ± 0.08	10.70 ^abc^ ± 0.03
	60	9.46 ^a^ ± 0.15	10.40 ^ab^ ± 0.07	10.43 ^ab^ ± 0.13	10.66 ^bcd^ ± 0.04
	75	9.60 ^a^ ± 0.00	10.48 ^ab^ ± 0.07	10.34 ^ab^ ± 0.04	10.62 ^cd^ ± 0.04
	90	9.56 ^a^ ± 0.07	10.34 ^ab^ ± 0.03	10.29 ^b^ ± 0.01	10.58 ^d^ ± 0.04

Means (±standard deviation) followed by different lowercase letters in the same column differ statistically from each other, according to the Tukey test (*p* ≤ 0.05). GS (100% S), GK1 (75% S, 25% kefir), GK2 (50% S, 50% kefir), GK3 (25% S, 75% kefir), GK (100% kefir). Average of the triplicate of two different processes.

**Table 3 foods-11-03148-t003:** Physicochemical analysis and caloric value of the five ice cream formulations (g/100 g).

Formulations	pH	Acidity %	Ashes	Protein %	Lipid %	Moisture %	Carbohydrates %	Kcal/100 g
GS	6.15 ^a^ ± 0.05	6.36 ^b^ ± 0.36	0.64 ^d^ ± 0.00	4.55 ^a^ ± 0.24	3.82 ^a^ ± 0.51	80.25 ^a^ ± 0.14	11.36 ^c^ ± 0.14	98.02 ^c^ ± 3.17
GK1	4.92 ^c^ ± 0.02	10.06 ^a^ ± 0.32	0.72 ^b^ ± 0.01	5.33 ^a^ ± 0.04	4.24 ^a^ ± 0.39	74.91 ^b^ ± 1.52	15.49 ^b^ ± 1.35	121.44 ^b^ ± 7.91
GK2	5.04 ^b^ ± 0.02	10.61 ^a^ ± 0.09	0.77 ^a^ ± 0.01	5.11 ^a^ ± 1.25	3.32 ^a^ ± 0.42	72.70 ^c^ ± 0.18	18.16 ^b^ ± 1.47	122.96 ^b^ ± 5.77
GK3	4.92 ^c^ ± 0.01	10.86 ^a^ ± 0.15	0.76 ^a^ ± 0.01	5.74 ^a^ ± 0.10	3.67 ^a^ ± 0.89	73.36 ^bc^ ± 0.36	17.20 ^b^ ± 1.08	124.79 ^b^ ± 3.72
GK	4.76 ^d^ ± 0.06	10.92 ^a^ ± 0.11	0.68 ^c^ ± 0.01	5.37 ^a^ ± 0.02	3.52 ^a^ ± 0.45	69.62 ^d^ ± 0.33	21.59 ^a^ ± 0.12	139.52 ^a^ ± 0.38

Means (±standard deviation) followed by different lowercase letters in the same column differ statistically from each other, according to the Tukey test (*p* ≤ 0.05). GS (100% S), GK1 (75% S, 25% kefir), GK2 (50% S, 50% kefir), GK3 (25% S, 75% kefir), GK (100% kefir). S = soy extract. Average of the triplicate of two different processes.

**Table 4 foods-11-03148-t004:** Total phenolic compounds and antioxidant capacity of ice cream after 0 and 90 days of storage.

Formulations	Total Phenolics (mgEAG/100 g)		Antioxidant Capacity (µmol Trolox/g)	
	T0	T90	T0	T90
GS	567.65 ^aD^ ± 35.60	271.00 ^bD^ ± 18.82	2.14 ^aB^ ± 0.46	1.80 ^aA^ ± 0.10
GK1	5970.93 ^aC^ ± 595.43	4755.51 ^bC^ ± 58.59	3.32 ^aA^ ± 0.05	1.84 ^bA^ ± 0.04
GK2	6783.94 ^aB^ ± 78.10	6221.41 ^bB^ ± 23.27	3.35 ^aA^ ± 0.02	1.95 ^bA^ ± 0.07
GK3	6670.40 ^aBC^ ± 32.63	6207.21 ^aB^ ± 19.74	2.81 ^aAB^ ± 0.20	1.93 ^bA^ ± 0.06
GK	7631.69 ^aA^ ± 47.73	7042.00 ^bA^ ± 48.34	2.99 ^aA^ ± 0.27	1.89 ^bA^ ± 0.05

Means (±standard deviation) followed by different lowercase letters on the same line differ statistically from each other, according to Test T (times—*p* ≤ 0.05). Means (±standard deviation) followed by different capital letters in the same column differ statistically from each other (formulations—*p* ≤ 0.05), according to the Tukey test (*p* ≤ 0.05). GS (100% S), GK1 (75% S, 25% kefir), GK2 (50% S, 50% kefir), GK3 (25% S, 75% kefir), GK (100% kefir). S = soy extract.

**Table 5 foods-11-03148-t005:** Quantification of anthocyanins cyanidin 3-glycoside and delphinidin 3-glycoside in ice cream after 0 and 90 days of storage.

Formulations	Cyanidin 3-Glycoside (mg/100 g)		Delphinidin 3-Glycoside (mg/100 g)	
	T0	T90	T0	T90
GS	0.29 ^aA^ ± 0.00	0.26 ^bA^ ± 0.00	11.11 ^aA^ ± 0.07	9.89 ^bA^ ± 0.00
GK1	0.23 ^aB^ ± 0.01	0.22 ^bB^ ± 0.01	8.77 ^aC^ ± 0.30	8.26 ^aB^ ± 0.00
GK2	0.22 ^aBC^ ± 0.00	0.18 ^bC^ ± 0.00	8.36 ^aC^ ± 0.07	7.07 ^bC^ ± 0.00
GK3	0.21 ^aC^ ± 0.00	0.16 ^bD^ ± 0.00	8.15 ^aC^ ± 0.10	6.95 ^bC^ ± 0.00
GK	0.19 ^aD^ ± 0.00	0.17 ^bD^ ± 0.00	10.23 ^aB^ ± 0.02	6.72 ^bC^ ± 0.00

Means (±standard deviation) followed by different lowercase letters on the same line differ statistically from each other, according to Test T (times—*p* ≤ 0.05). Means (±standard deviation) followed by different capital letters in the same column differ statistically from each other (formulations—*p* ≤ 0.05), according to the Tukey test (*p* ≤ 0.05). GK1 (75% S, 25% kefir), GK2 (50% S, 50% kefir), GK3 (25% S, 75% kefir), GK (100% kefir). S = soy extract.

**Table 6 foods-11-03148-t006:** Instrumental color parameters (L *, a *, b *, C * and H°) of ice creams after 0 and 90 days of storage.

		Formulations				
		GS	GK1	GK2	GK3	GK
L *	T 0	0.60 ^cA^ ± 0.12	0.84 ^bA^ ± 0.07	0.47 ^cA^ ± 0.08	0.91 ^bA^ ± 0.07	1.21 ^aA^ ± 0.03
	T 90	0.59 ^bA^ ± 0.10	0.26 ^cB^ ± 0.22	0.32 ^bcB^ ± 0.06	1.02 ^aA^ ± 0.05	0.94 ^aB^ ± 0.16
a *	T 0	0.15 ^bA^ ± 0.05	0.22 ^abA^ ± 0.07	0.15 ^bA^ ± 0.05	0.29 ^abA^ ± 0.04	0.36 ^aA^ ± 0.10
	T 90	0.16 ^abA^ ± 0.06	0.05 ^bB^ ± 0.01	0.09 ^bA^ ± 0.05	0.20 ^abA^ ± 0.14	0.27 ^aA^ ± 0.04
b *	T 0	0.27 ^abA^ ± 0.07	0.33 ^abA^ ± 0.02	0.33 ^abA^ ± 0.02	0.25 ^bA^ ± 0.08	0.40 ^aA^ ± 0.02
	T 90	0.22 ^aA^ ± 0.04	0.25 ^aB^ ± 0.02	0.11 ^bB^ ± 0.03	0.23 ^aA^ ± 0.08	0.25 ^aB^ ± 0.04
C *	T 0	0.31 ^bA^ ± 0.07	0.40 ^abA^ ± 0.04	0.36 ^bA^ ± 0.01	0.38 ^abA^ ± 0.08	0.40 ^aA^ ± 0.02
	T 90	0.28 ^aA^ ± 0.05	0.26 ^abB^ ± 0.02	0.15 ^bB^ ± 0.03	0.25 ^abA^ ± 0.11	0.25 ^aB^ ± 0.04
h°	T 0	60.78 ^a^^b^^A^ ± 6.99	56.50 ^abB^ ± 7.97	65.47 ^aA^ ± 8.67	40.00 ^bA^ ± 7.50	40.38 ^bA^ ± 7.79
	T 90	54.32 ^ab^^A^ ± 11.38	78.71 ^aA^ ± 3.18	41.35 ^bB^ ± 8.06	53.50 ^a^^bA^ ± 14.02	40.50 ^bA^ ± 3.79

Means (±standard deviation) followed by different lowercase letters in the same row (sample comparison) and different uppercase letters in the same column (time comparison) differ statistically according to the Tukey’s test (*p* ≤ 0.05). GK1 (75% S, 25% kefir), GK2 (50% S, 50% kefir), GK3 (25% S, 75% kefir), GK (100% kefir). S = soy extract.

**Table 7 foods-11-03148-t007:** Ice cream acceptance at T0 and T90 days of storage.

Time	Formulations				
	GS	GK1	GK2	GK3	GK
T0					
Appearance	6.73 ^aA^ ± 1.66	6.88 ^aA^ ± 1.67	7.09 ^aA^ ± 1.60	7.21 ^aA^ ± 1.37	7.17 ^aA^ ± 1.58
Color	6.75 ^aA^ ± 1.64	6.79 ^aA^ ± 1.56	7.02 ^aA^ ± 1.50	7.23 ^aA^ ± 1.36	7.18 ^aA^ ± 1.48
Aroma	5.73 ^aA^ ± 1.62	5.63 ^aA^ ± 1.56	5.98 ^aA^ ± 1.44	5.90 ^aA^ ± 1.40	5.95 ^aA^ ± 1.67
Flavor	5.57 ^abA^ ± 2.05	5.18 ^bA^ ± 1.98	5.92 ^aA^ ± 1.85	5.84 ^abA^ ± 1.91	6.06 ^aB^ ± 2.12
T90					
Appearance	6.31 ^bA^ ± 1.86	6.28 ^bB^ ± 1.94	6.56 ^abB^ ± 1.67	6.69 ^abB^ ± 1.64	7.15 ^aA^ ± 1.59
Color	6.22 ^bB^ ± 1.87	6.24 ^bB^ ± 1.83	6.55 ^abB^ ± 1.71	6.59 ^abB^ ± 1.71	7.03 ^aA^ ± 1.62
Aroma	5.90 ^aA^ ± 1.71	5.85 ^aA^ ± 1.51	5.84 ^aA^ ± 1.47	6.10 ^aA^ ± 1.54	6.29 ^aA^ ± 1.65
Flavor	5.87 ^bA^ ± 1.98	5.02 ^cA^ ± 1.88	5.47 ^bcA^ ± 1.83	5.95 ^bA^ ± 1.70	6.86 ^aA^ ± 1.94

Means (±standard deviation) followed by different lowercase letters on the same line differ statistically from each other, and different uppercase letters on the same column statistically according to the Tukey test (*p* ≤ 0.05). GK1 (75% S, 25% kefir), GK2 (50% S, 50% kefir), GK3 (25% S, 75% kefir), GK (100% kefir). S = soy extract.

**Table 8 foods-11-03148-t008:** Frequency of sensory attributes and Cochran’s Q test associated with each ice cream at the beginning of the storage period (T0).

	Formulations					
Attributes	GS	GK1	GK2	GK3	GK	*p*
Sweet	24	32	32	30	53	<0.0001
Milk flavor	22	19	30	26	16	0.065
Artificial jaboticaba flavor	20	19	19	23	23	0.822
Natural jaboticaba flavor	24	25	25	33	38	0.032
Creamy texture	48	51	49	43	58	0.299
Soft texture	48	39	44	41	40	0.680
Mild flavor	42	36	49	41	40	0.413
Watered down	23	31	29	25	19	0.243
Acid taste	04	05	14	18	26	<0.0001
Refreshing flavor	31	23	31	38	44	0.005
Vanilla flavor	17	19	19	20	18	0.970
Fruit scent	22	18	13	22	23	0.240
Aftertaste	47	46	32	38	35	0.062
Fermented flavor	26	28	24	29	27	0.890
Artificial scent	10	14	12	09	10	0.695
Soy flavor	77	69	57	58	55	0.001
Roughness	37	38	31	40	35	0.649
Acid scent	03	03	05	08	05	0.380
Fermented scent	24	21	21	25	26	0.808
Yogurt flavor	21	20	30	25	36	0.033
Astringency	15	18	20	17	20	0.791
Purple fruit color (berries)	44	42	42	39	45	0.805
Natural jaboticaba color	39	38	39	48	42	0.218
Attractive color	42	37	47	47	50	0.138

n = 24. *p* ≤ 0.05 indicates a significant difference in Cochran’s Q test. GK1 (75% S, 25% kefir), GK2 (50% S, 50% kefir), GK3 (25% S, 75% kefir), GK (100% kefir) S = soy extract.

**Table 9 foods-11-03148-t009:** Frequency of sensory attributes and Cochran’s Q test associated with each ice cream at the end of the storage period (T90).

Attributes	Formulations					
	GS	GK1	GK2	GK3	GK	*p*
Sweet	50	26	28	40	75	<0.0001
Milk flavor	29	25	21	19	22	0.369
Artificial jaboticaba flavor	22	22	25	26	19	0.702
Natural jaboticaba flavor	32	25	31	43	50	<0.0001
Creamy texture	76	51	43	51	72	<0.0001
Soft texture	53	40	43	38	47	0.182
Mild flavor	56	39	47	45	49	0.186
Watered down	15	39	36	38	16	<0.0001
Acid taste	03	10	10	22	24	<0.0001
Refreshing flavor	28	19	27	38	52	<0.0001
Vanilla flavor	28	27	27	21	24	0.694
Fruit scent	23	16	22	30	38	0.001
Aftertaste	39	44	45	40	30	0.115
Fermented flavor	20	27	23	35	29	0.081
Artificial scent	18	19	16	12	11	0.238
Soy flavor	57	67	65	43	34	<0.0001
Roughness	31	41	44	49	37	0.044
Acid scent	03	06	05	09	09	0.225
Fermented scent	15	10	17	19	20	0.192
Yogurt flavor	16	17	27	29	39	0.000
Astringency	17	32	28	32	18	0.001
Purple fruit color (berries)	47	41	57	48	56	0.002
Natural jaboticaba color	39	37	42	41	45	0.590
Attractive color	26	29	37	44	53	<0.0001

n = 24. *p* ≤ 0.05 indicates a significant difference in Cochran’s Q test.GK1 (75% S, 25% kefir), GK2 (50% S, 50% kefir), GK3 (25% S, 75% kefir), GK (100% kefir). S = soy extract.

## Data Availability

The datasets generated during analyzed during the current study are available from the corresponding author on request.

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
