# Peer review of "Vegan Ice Cream Made from Soy Extract, Soy Kefir and Jaboticaba Peel: Antioxidant Capacity and Sensory Profile"

_foods, 2022, doi:10.3390/foods11193148_

Round 1

Reviewer 1 Report

The weakness :

1)      Absence of control sample: cow milk-based ice cream

2)      Absence of discussion about the relation between physico-chemical properties changes and sensorial profile changes during the storage.

Other comments:

L31-35. The sentence is too long, it should be split out.

L47-50. The sentence is too long, it should be split out.

L59: the textural properties, flavor, color, and product’s acceptance.

L66. was obtained from

L69. (Dupont, …). (full stop)

L75. Provide the year of harvest

.. sanitized in aqueous sodium chloride solution (100 mg/L) for 30 minutes ….

L77. … in ovens equipped with air circulation at 60°C to a constant moisture content …

L83-86. Rewrite the sentence. It is not clear. The composition also should be clear out that it is based on dry basis or wet basis.

Table 1. Specify if the formulation is based on dry basis or wet basis

L128. Please use the uniform citation style.

L131: Delete ‘enough’

L201. … and had coliform

L232. What is the conclusion about your products regarding ISAPP regulation?

Table 2. The data set of GS is missing.

Table footnote. …. GK3 (…). (full stop).

Table 3. Use a uniform police size

Table footnote. …. GK3 (…). (full stop).

L268. EHS should be written in full term.

L271. .. samples containing kefir

Table 4. Use a uniform police size

Table footnote. …. GK3 (…). (full stop).

Table 5. Use a uniform police size

Table footnote. …. GK3 (…). (full stop).

Line 309: … of anthocyanins (Tables 3,5), ….

L315-L320. State briefly how the evolution of colour during the storage for formulations.

Table 6. Use a uniform police size

Table footnote. …. GK3 (…). (full stop).

Line 344. … regarding flavor and appearance???

Table 7. Use a uniform police size

Table footnote. …. GK3 (…). (full stop).

Improve the table format, be more readable

L349-351. Please cite a table

L354.. with those obtained in the hedonic test for storage time of T0???

L361-362: the p for ‘after taste’ is > 0.05. the aftertaste should be replaced by acid taste and yoghurt flavor.

Table 8. Table footnote. …. GK3 (…). (full stop).

L370-372. Several attributes are missing, such as astringency, watered down, acid taste..

Table 9. Table footnote. …. GK3 (…). (full stop).

L390. The attribute of attractive color belongs to GK sample.

L407. The best anthocyanin profile belongs to GS sample.

Author Response

Por favor, verifique o anexo

Reviewer 2 Report

The manuscript is interesting. The design is good but it should be thoroughly revised Tables need attention and Reference section should be according to the journals guideline

Round 2

Reviewer 1 Report

The manuscript was revised correctly and, therefore, can be published in this present form